# X-ray crystallography reveals molecular recognition mechanism for sugar binding in a melibiose transporter MelB

Lan Guan [1✉] & Parameswaran Hariharan [1]

Major facilitator superfamily_2 transporters are widely found from bacteria to mammals. The melibiose transporter MelB, which catalyzes melibiose symport with either $Na^+$, $Li^+$, or $H^+$, is a prototype of the $Na^+$-coupled MFS transporters, but its sugar recognition mechanism has been a long-unsolved puzzle. Two high-resolution X-ray crystal structures of a *Salmonella typhimurium* MelB mutant with a bound ligand, either nitrophenyl-α-D-galactoside or dodecyl-β-D-melibioside, were refined to a resolution of 3.05 or 3.15 Å, respectively. In the substrate-binding site, the interaction of both galactosyl moieties on the two ligands with MelB$_{St}$ are virtually same, so the sugar specificity determinant pocket can be recognized, and hence the molecular recognition mechanism for sugar binding in MelB has been deciphered. The conserved cation-binding pocket is also proposed, which directly connects to the sugar specificity pocket. These key structural findings have laid a solid foundation for our understanding of the cooperative binding and symport mechanisms in $Na^+$-coupled MFS transporters, including eukaryotic transporters such as MFSD2A.

[1] Department of Cell Physiology and Molecular Biophysics, Center for Membrane Protein Research, School of Medicine, Texas Tech University Health Sciences Center, Lubbock, TX, USA. ✉email: Lan.Guan@ttuhsc.edu

Membrane transporters play critical roles in human health and disease, and they are potential drug targets with increasing interests in pharmaceutical applications. While active transporters can move specific solutes into cells effectively, a majority of available drugs enter cells by inefficient solubility diffusion. This is largely due to lack of knowledge on active transporters, in particular the detailed understanding about substrate-binding sites and transport mechanisms. The glycoside–pentoside–hexuronide:cation symporter family (GPH)[1], which is a subgroup of the major facilitator superfamily (MFS) of membrane transporters found from bacteria to mammals that catalyze a coupled substrate transport with monovalent Na[+], Li[+], or H[+], Na[+] or Li[+], H[+] or Li[+], or only H[+] (ref. [2]). Most members transport glycosides, but the substrates transported by this family are diverse from small molecules, such as glucose, melibiose, raffinose, or sulfoquinovose, to bigger molecules such as phospholipids. For example, the Na[+]-coupled MFSD2A in blood-brain barrier in the brain and blood-retinal barrier in eyes mediates the delivery of the essential omega-3 fatty acids for neurodevelopment[3–5]. The human MFSD13A transporter, which is expressed in many tissues but overexpressed in colorectal cancer cells, has been successfully targeted for diagnosis and treatment[6,7]. Based on the larger Pfam database, these proteins belong to MFS_2 family[8] with greater than 11,000 sequenced genes across thousands species (https://pfam.xfam.org/family/PF13347#tabview=tab0). The first high-resolution crystal structure of this large family was obtained from a bacterial melibiose permease MelB[9]. Lack of high-resolution substrate-bound 3-D structure is a bottleneck for unraveling molecular recognition and transport mechanisms, which significantly hampered the developments towards the pharmaceutical applications as drug targets or as vehicles for drug delivery.

The bacterial melibiose permease MelB is one of the few membrane transporters discovered at earlier years[10], and it has been serving as a model system for the study of cation-coupled transport mechanisms[11–14], as well as the development of novel analytical techniques for membrane protein research[13,15–19]. It is a well-documented representative from MFS_2 family[1,9,12,14,20–24]. MelB catalyzes the stoichiometric galactose or galactosides symport with monovalent cation either Na[+], Li[+], or H[+], with no affinity for glucose or glucosides[25–28]. The symport reactions are reversible, and the polarity of transport is determined by the net electrochemical gradients from both substrates. In either symport process, the coupling between the driving cation and driven sugar is obligatory[11,27,29,30]. The previous X-ray 3-D crystal structure of the MelB of *Salmonella typhimurium* (MelB$_{St}$) shows a MFS fold at an outward-facing conformation[9]; while sugar binding or cation binding was not resolved, this structure showed that the residues that have been functionally determined to be important for the co-substrate binding are located within a large cavity[9,11,31–38]. Like other MFS transporters[39–44], the well-recognized alternating-access process has been also proposed in MelB[23,31,36].

The alternating-access process only describes the conformational changes during transport by alternatively opening and closing the access path to the ligand-binding site on both sides, allowing the solute from one side of the membrane to the other side. However, it is often not clear how a protein responds to its specific substrate binding and initiates the alternating-access process. Recently, the binding of Na[+] and melibiose to MelB$_{St}$ has been characterized by the isothermal titration calorimetry (ITC)[24]. Combined findings from analysis of the binding free energy and heat capacity changes in the thermodynamic cycle, it has been clear now that the core mechanism for this symporter is the positive cooperativity of the co-substrates (melibiose and Na[+]) to ensure obligatory coupling and overcome energetic barriers[24,45].

To further gain the structural basis, in this study, we determined two galactoside-bound crystal structures of an uniport mutant D59C MelB$_{St}$ that carries a mutation at a highly conserved cation site. Asp59 on helix II was well-studied in MelB of *Escherichia coli* (MelB$_{Ec}$)[36,46,47]. The D59C mutation in both MelB$_{Ec}$ and MelB$_{St}$ abolishes Na[+] binding and eliminates the stimulation of Na[+] on galactoside binding or transport, but they can catalyze the melibiose transport independent of cations. It has been also suggested that Asp59 is the ligand for Na[+] and also for H[+] and D59C mutant behaves like a uniporter[9,24,36,46,47]. The two crystal structures of D59C MelB$_{St}$ revealed the binding of the melibiose analog 4-nitrophenyl-α-D-galactopyranoside (α-NPG) or the novel melibiose-derived detergent n-dodecyl-β-melibioside (DDMB). Both structures are virtually identical; each exhibits an open periplasmic vestibule with no bound cation, representing the binary complex of MelB with a bound galactopyranoside at an outward-facing conformation.

## Results

**Melibiose transport and ligand binding.** [³H]Melibiose transport assays with intact *E. coli* cells (*melB*⁻) confirmed that the D59C MelB$_{St}$ mutant lost all three modes of melibiose active transport coupled to H[+], Na[+], or Li[+] (ref. [9]), even with increased concentration of Na[+] or Li[+] (Fig. 1a). Interestingly, this mutant showed a normal melibiose fermentation with red colonies on MacConkey agar plates containing melibiose as the sole carbohydrate source (Fig. 1a insets). This transport rate-dependent color development assay indicates that melibiose translocation pathway is retained with this mutant. When placing Ala residue at the position Asp59, the D59A mutant behaves similar to the D59C mutant.

ITC measurements with MelB$_{St}$ in solution were used to analyze the co-substrates binding (Supplementary Fig. 1; Table 1). We have reported the cooperative binding between Na[+] and melibiose in WT MelB$_{St}$[24], and recently we have also showed that cooperative binding acts as a regulatory mechanism to ensure the obligatory coupling between these two substrates[45]. The D59C mutant at apo state exhibits a $K_d$ value for melibiose with a surprisingly 2-fold lower than that of WT; the effect of Na[+] or Li[+] on the melibiose binding was not detected. Consistently, α-NPG also binds to apo D59C mutant with about 4-fold increased affinity, again with no Na[+] or Li[+] effect. Li[+] binding measurements to the WT showed a $K_d$ value at apo state of 0.40 ± 0.10 mM, which was decreased by 6-fold in the presence of melibiose; notably, the cooperativity is less than melibiose with Na[+] with a 8-fold change (Supplementary Fig. 1; Table 1)[24]. For D59C mutant, all heat changes derived from Na[+] or Li[+] binding are quite small even with increased concentration or in the presence of melibiose, which prevents an accurate determination of the dramatically elevated $K_d$ values. Collectively, D59C MelB$_{St}$ selectively eliminates the cation binding and cation stimulation on melibiose binding, as well as their co-transport, but retains the ability of melibiose binding and translocation. Thus, D59C MelB$_{St}$ mutant is an uncoupled mutant as proposed for MelB$_{Ec}$[24].

DDMB is a melibiose-derived novel detergent containing a melibiosyl moiety and a 12-carbon chain[48] (Fig. 1b). A well-established FRET measurement[28,35], which is based on MelB Trp residues → dansyl moiety on the bound fluorescent analog dansyl-2-galactoside (D²G) FRET, was used to determine MelB$_{St}$ affinity for DDMB with purified protein samples in solutions. Addition of DDMB or α-NPG into the D²G-bound WT MelB$_{St}$ solution leaded to a large decrease in fluorescent intensity as a result from the exchange of MelB$_{St}$-bound D²G by DDMB

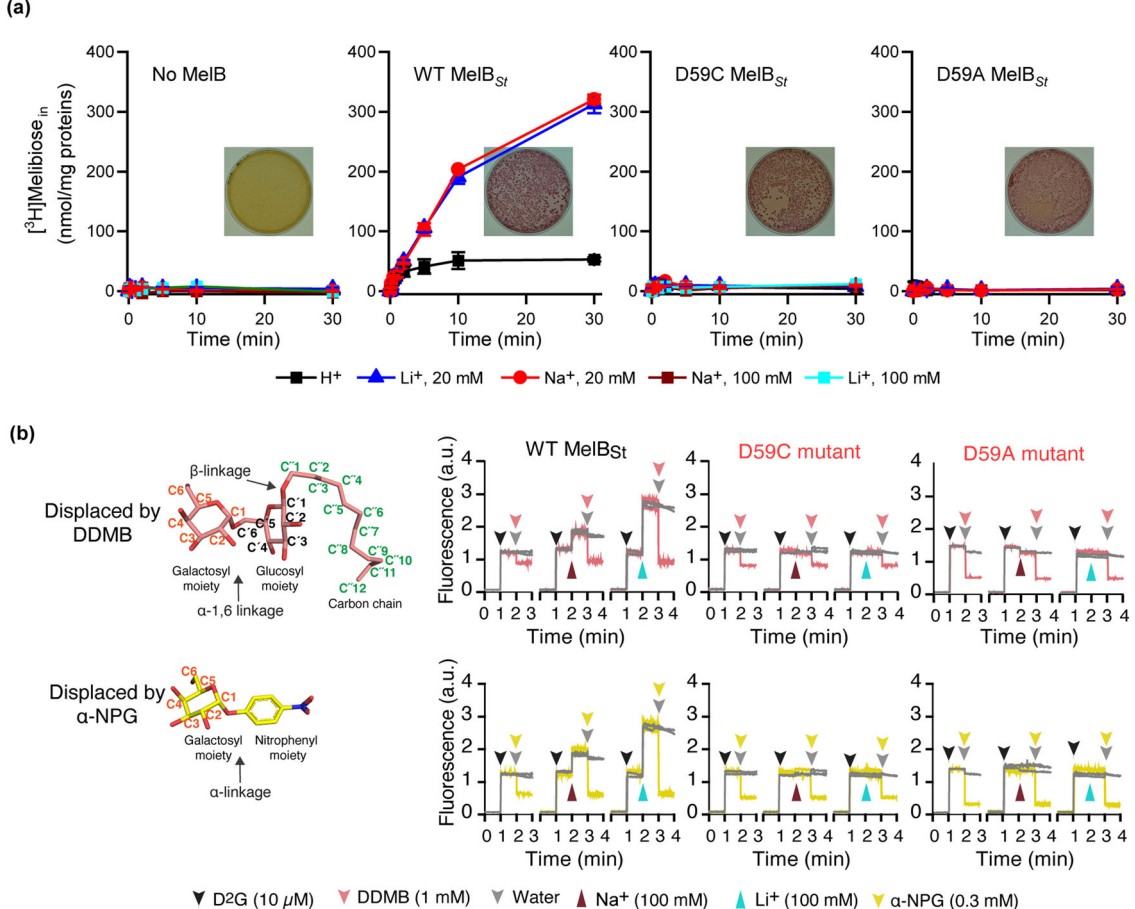

**Fig. 1 Functional characterizations. a** [³H] Melibiose transport time course with intact cells. *E. coli* DW2 strain (*melB⁻*, *lacZ⁻Y⁻*) expressed with the WT, D59C, or D59A MelB$_{St}$ mutants were subjected to [³H]melibiose active transport assay at 0.4 mM and 10 mCi/mmol in the presence of 20 mM or 100 mM NaCl or LiCl by a fast filtration method. Inset, melibiose fermentation assay. Transformants with DW2 strain were plated on MacConkey agar plates containing 30 mM melibiose as the sole carbohydrate source and pH sensor neutral red, and incubated at 37 °C for 16–18 h before imaging. Number of tests = 3, from different batch of cells or duplicate measurements from same batch of sample. **b** Ligand binding by Trp→dansyl galactoside (D²G) FRET assay. Purified proteins at 1 μM concentration in 20 mM Tris-HCl, pH 7.5, 100 mM CholCl, 10% glycerol, and 0.03% UDM were subjected to steady-state fluorescence measurements as described in Methods. On the time-trace, 1.8 μL of 1 mM D²G at 10 μM was added into 180 μL of protein solutions at 1-min point, followed by supplying 10 μL of 5 mM DDMB or 5 mM α-NPG to displace the bound D²G. For testing Na⁺ or Li⁺ binding, they are added into the solution before displacement with the second sugar. The structure of DDMB or α-NPG is shown in a chair form, and the carbon positions are labeled. Glycosidic bonds are indicated by arrows. Both are α-galactosides containing a substituent at the anomeric carbon-1. The data were from duplicate measurements.

**Table 1 $K_d$ of substrate binding to MelB$_{St}$.**

|  | Conditions | Melibiose (mM) | α-NPG (μM) | Conditions | Na⁺ binding (mM) | Li⁺ binding (mM) |
|---|---|---|---|---|---|---|
| WT | **Apo** | 9.28 ± 0.23[a,b] (n = 3) | 43.20 ± 0.62 (n = 2) | **Apo** | 0.68 ± 0.03[b] (n = 4) | 0.40 ± 0.10 (n = 2) |
|  | **Na⁺ (100 mM)** | 1.09 ± 0.06[b] (n = 4) | 25.69 ± 4.42 (n = 2) | **Melibiose (50 mM)** | 0.08 ± 0.01[b] (n = 4) | 0.07 ± 0.01 (n = 2) |
| D59C | **Apo** | 4.66 ± 0.10 (n = 3) | 10.47 ± 1.34 (n = 2) | **Apo** | /[c] (n = 2) | / (n = 2) |
|  | **Na⁺ (100 mM)** | 4.96 ± 0.11 (n = 3) | 11.97 ± 0.09 (n = 2) | **Melibiose (50 mM)** | / (n = 1) | / (n = 1) |

*n*, the number of tests. The protein conditions for each individual test were in bold.
[a]SE, Standard error.
[b]Data from the publication in ref. [22] or averaged with more tests.
[c]No curve fitting was carried out.

(Fig. 1b; pink) or α-NPG (Fig. 1b; yellow). The data strongly indicate that DDMB can competitively bind to WT MelB$_{St}$. Na⁺ or Li⁺ stimulation on the FRET with WT is typically indicated as the increase in fluorescent intensity upon adding NaCl or LiCl (at the 2-min time point) prior to the displacement. The data clearly show that positive cooperativity with Na⁺ or Li⁺ also exists with DDMB, although it is not possible to obtain information about

the effect of DDMB on the affinity of Na⁺ or Li⁺ from this method. The D59A mutant also binds α-NPG and DDMB, but with no Na⁺ or Li⁺ stimulation, which is expected (Fig. 1b).

Quantitatively, the half-maximal concentration for displacing the bound D²G (IC$_{50}$) was determined for DDMB and α-NPG (Supplementary Fig. 2). Titration of DDMB was carried out at a final concentration less than the critical micelle concentration

**Table 2 IC$_{50}$ of galactoside displacement on bound D$^2$G by MelB$_{St}$.**

| | DDMB (μM)[a] | | α-NPG (μM) | |
| | Apo | Na$^+$ (100 mM) | Apo | Na$^+$ (100 mM) |
|---|---|---|---|---|
| WT | 26.13 ± 5.46 (n = 3) | 4.79 ± 0.28 (n = 3) | 58.17 ± 11.53[b] (n = 2) | 22.96 ± 2.24 (n = 3) |
| D59C | 20.16 ± 7.37 (n = 2) | 23.56 ± 3.90 (n = 2) | 22.15 ± 5.75 (n = 2) | 21.27 ± 1.67 (n = 2) |

[a]IC50 value could likely be over-estimated as explained in the text.
[b]SE, standard error; n, the number of tests.

(CMC, 295 μM) to keep DDMB in monomeric form, and the FRET intensity was decreased along with the increase in DDMB concentrations. As a negative control, the maltoside-based detergents DDM (CMC, 170 μM) and UDM (CMC, 590 μM) were tested under the same conditions, which only showed a dilution effect similar to the addition of water, and supports the specific effect observed from DDMB. The curve fitting reveals IC$_{50}$ value of 26.13 ± 5.46 μM with the apo WT MelB$_{St}$, which is decreased by 5-fold to 4.79 ± 0.28 μM in the presence of Na$^+$ (Table 2; Supplementary Fig. 2).

Since both monomer and micelles co-exist in the concentrated solutions used for the titration and detergent CMC is also affected by buffer conditions, the true concentration of the effective monomeric form cannot be estimated. Thus, the IC$_{50}$ might be lower than the determined values. Even so, DDMB, as a ligand for MelB$_{St}$, showed the highest affinity of all the galactosides tested. The D59C mutant yields a IC$_{50}$ of 20 μM for DDMB; consistently, there is little effect by Na$^+$. Importantly, use of this high-affinity substrate further confirms the uncoupled property with this mutant, as well as defines the positive cooperativity based on the specific galactosyl moiety with selected cations.

**X-ray crystal structures of D59C MelB$_{St}$ with bound galactoside.** Purified D59C MelB$_{St}$ in UDM solution exhibits an improved thermostability with a melting temperature ($T_m$) of 60 °C as detected by circular dichroism (CD) spectroscopy, which is greater by 6 °C than the WT (Supplementary Fig. 3)[49]. Both proteins are well-folded with α-helical secondary structure overwhelmingly dominated. Two crystal structures of D59C MelB$_{St}$ with a bound α-NPG or DDMB were modeled from positions 2 to 454 without a gap (Fig. 2a), refined to a resolution of 3.05 Å and 3.15 Å, and assigned with a PDB access identification code under 7L17 and 7L16, respectively. The phase was determined by MR-SAD based on selenomethionine (SeMet)-incorporated D59C MelB$_{St}$ and a model from structure pdb id, 4m64[9] (Supplementary Fig. 4a, b). The two native structures are superimposed very well, with rmsd of 0.187 Å, showing a typical MFS fold (Fig. 2a) with 12 transmembrane helices. The cytoplasmic middle loop carries a short α-helix linking the two helical bundles formed by the N- and C-terminal six α-helices; the C-terminal tail also contains a short α-helix; the last 26 residues were unassigned due to poor electron density.

An open cavity at periplasmic side is formed by 8 inner-layer helices (I, II, IV, V, VII, VIII, X, and XI) (Supplementary Fig. 5); at the apex of the cavity in both structures, one molecule of α-NPG (Fig. 2a) or DDMB (Fig. 2b) is bound to the sugar-binding pocket as indicated. This pocket is opened to the periplasmic surface through the solvent-accessible periplasmic vestibule, allowing sugar to enter the sugar-binding site. Overlay of the two structures shows a narrow pocket that only hosts the specific galactosyl moiety of both ligands as indicated as black cycle in panel b, named sugar specificity determinant pocket. The cavity inside the red circle is big enough to accommodate a detergent tail with 12 carbons, which is called a non-specific binding pocket.

More details will be discussed in the following sections. There is no solvent-accessible tunnel towards the cytoplasmic surface. The cytoplasmic tail together with helices IV-loop-V, X-loop-XI, as well as middle loop, stabilize the binary complex of MelB$_{St}$ with a bound galactoside with closed inside and open outside.

**α-NPG binding site.** As shown in Fig. 3a, a strong electron density blob links the N-terminal helices I and IV, which fits well to a α-NPG molecule that was presented in the crystallization drop. The calculated unbiased composite omit map or polder map also show the orientation and interactions of α-NPG (Supplementary Fig. 6a). The galactosyl moiety is located in a small pocket and form multiple close contacts with the N-terminal helices (I, IV, and V), and has little interaction with the C-terminal helices (X and XI) (Fig. 3b). The substituent nitrophenyl group is exposed to the large solvent-filled cavity, with little polar interaction with the protein. In the cavity, there are some unassigned densities that also presents in the DDMB-bound structure. The molecular recognition of α-NPG by MelB highly focuses around the specific galactosyl moiety. Each of the four hydroxyl groups (OH) are fully liganded by 2–3 hydrogen-bonds (H-bond, mainly bifurcated H-bonds), and the galactosyl ring is sandwiched by aromatic sidechains via Ch-π interactions. The two negatively charged Asp19 (helix I) and Asp124 (helix IV) make the major contributions to the four OH groups (Fig. 3a, b). Asp19 and Asp124 accept hydrogen atoms from C2- and C3-OH, or C4-OH and C6-OH of the galactosyl moiety, respectively, at distances less than 3.5 Å. Notably, the structure resolution was limited to 3.05 Å; at this resolution, the interatomic distance is still an estimation. The distance measurement will be used to describe the relative orientation and define the H-bonding interactions. Asp124, at a H-bond distance to Trp128 on the same helix, also forms a salt-bridge interaction with Lys18 (helix I), and this interaction supports Lys18 residue to donate a H to C3-OH and stabilizes this H-bond donor at a position between the two H-bond acceptors Asp19 & Asp124 residues. Furthermore, Arg149 donates two H bonds to C2-OH; Trp128 also donates a bifurcated H-bond to C3- and C4-OH, in addition to Asp124. From the C-terminal helical bundle, Lys377 and Gln372 (helix XI) are within 5 Å distance from C6-OH. Tyr120 (helix IV) forms a H-bond with Lys18 with no close contact to the sugar (>3.5 Å distance); Thr373 (helix XI) forms a H-bond with Asp124. Characteristically, all these interactions link together around the galactosyl moiety, and form a widespread salt-bridge assisted H-bonding network to "hang" the sugar in the middle of protein (Figs. 1, 3c).

Four aromatic residues are present in this sugar-binding cavity (Fig. 3b). Trp128 and Tyr120 were engaged in this H-bonding network. Trp342 (helix X) shapes the binding pocket by providing Ch-π interactions with both galactosyl and phenyl rings, and it also form aromatic stacking with Trp128. Tyr26 (helix I) forms an aromatic stacking interaction with the phenyl ring of α-NPG and indole ring of Trp116 (helix IV), and it could also form a potential H-bond with Lys377. Ile22, Trp26, Trp116, and Ala152 (Helix V) are located on the relatively hydrophobic

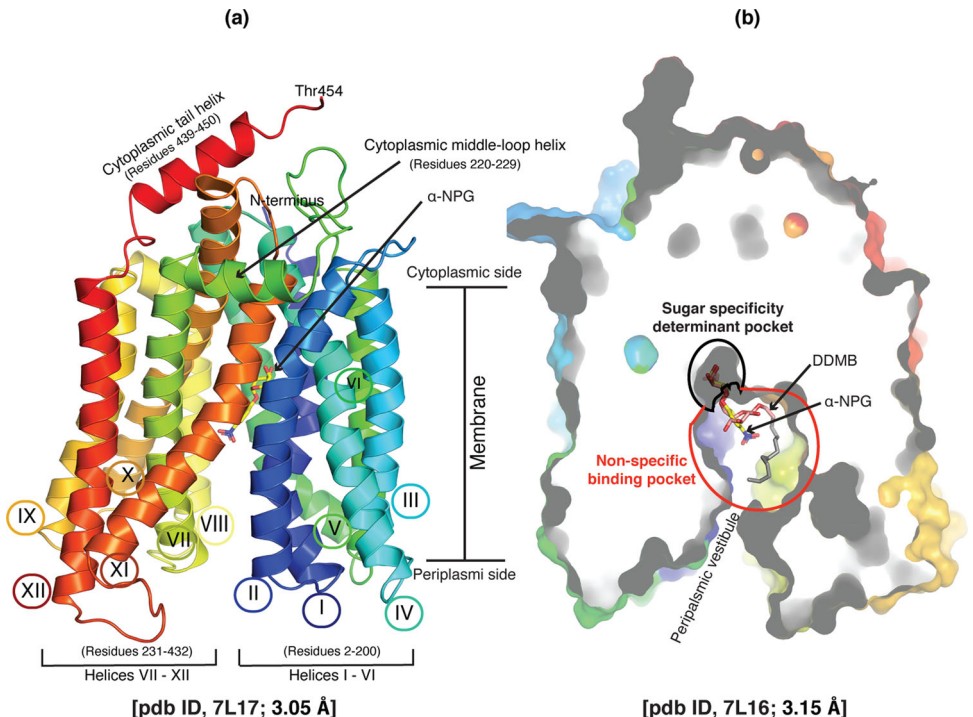

**Fig. 2 X-ray crystal structures of D59C MelB_St with bound α-NPG or DDMB. a** Overall fold and helical packing [PDB ID, 7L17]. The D59C MelB_St structure with bound α-NPG is viewed parallel to the membrane and shown in cartoon representation, which is colored in rainbow from N-terminus in blue to C-terminal end in red. Both termini are located in the cytoplasmic side as indicated. The membrane-spanning region is proximately estimated and indicated. Transmembrane helix is numbered in Roman numerals, and the two cytoplasmic helices is named as cytoplasmic middle-loop helix or cytoplasmic tail helix. The amino-acid sequence positions for the N-terminal 6 helices (I-VI) and the C-terminal 6 helices (VII-XII), as well as the two cytoplasmic helices, are indicated. The bound α-NPG is indicated. **b** Outward-facing conformation with bound DDMB [PDB ID, 7L16]. The structure is shown in a cross-section of surface representation with the N-terminal bundle orientated at the left side. DDMB is shown in pink and tail (C2-12) in gray. The sugar specificity determinant pocket is indicated by a black cycle, and the non-specific binding pocket is indicated by a red cycle. A α-NPG molecule was included that was from the α-NPG-bound structure [7L17] after overlayed with the DDMB-bound structure. PDB identification code for both structures are presented.

face of the galactosyl ring; Trp342 and Val376 (helix XI) are located on the other face; these residues provide a hydrophobic environment favored by the phenyl ring. In addition, Asn251, at 4.5 Å distance to the nitro group in the *para* position, together with further distancing Asn248 and Asn244 on the same face of helix VII, they may contribute a polar environment for this nitro group. These chemical environments around the nitrophenyl moiety could explain why α-NPG exhibits 100-fold greater affinity compared to the native substrate melibiose (Table 1). 

Mutations on most of these positions largely reduce the binding or transport in MelB_St and MelB_Ec, and some mutants also affected affinity to both sugar and cation[9,36,50–53].

**DDMB-binding site**. The DDMB-bound structure of D59C mutant is indistinguishable from that with α-NPG, but the electron density blob in the galactoside-binding pocket differs from that for α-NPG. It fits fairly well with a melibiosyl moiety of DDMB as verified by the composite omit map and polder map (Fig. 4a, Supplementary Fig. 6b, d), although the density is much weaker than that for α-NPG. Notably, the well-defined β-glycosidic bond on C1′ position supports the orientation of the galactosyl moiety in the sugar specificity site. The C3-12 carbon chain tail exhibits weak or no electron density with an occupancy set to 0 or 0.3 and colored in gray. Overall, the galactosyl moiety binds to the protein virtually indistinguishable from that of α-NPG. The glucosyl ring on melibiosyl moiety is nearly perpendicular to the galactosyl ring, which makes a H-bond with Trp342 on the C1′-O atom (Fig. 4b). In addition, the polar residues

Asn244 (helix VII), Ser153 (helix V), and Thr338 (helix X) are at distances of <6.0 Å to the glucosyl moiety, which could contribute a polar environment. This binding interaction of melibiosyl moiety may provide clues how the native substrate melibiose binds to MelB_St. 

Superposition of the two structures clearly reveals that the binding residues recognizing the galactosyl moiety are located on the five helices of the two helical bundles, and they form a narrow sugar specificity determinant pocket (Figs. 2b and 3c). Within the pocket, a salt-bridge-assisted H-bond network and Ch-π interactions play critical roles in the recognition of galactosides. This sugar-binding site is characteristic of a narrow specificity pocket with a large non-specific binding-cavity, which might indicate how other homologs recognize their different substrates.

**Proposed cation-binding pocket**. Asp55 and Asp59 on helix II have been functionally determined to be the Na$^+$-binding residues, which dictate the Na$^+$ coordination[24,36,46,47,54]. In both D59C mutant structures, position-59 is clearly not in direct contact with sugar, but the D59C sidechain is only 6.9-Å away from the C6-OH on the specific galactosyl moiety (Fig. 5a). Compared to the one helical-turn apart Asp55, which is in salt-bridge interaction with Lys377 (helix XI), the position-59 is less solvent accessible (Fig. 5b). The side chain on Lys377 is only resolved to Cε position due to poor electron density. It could form an interaction with Asp55 by a salt-bridge or with C6-OH on the sugar by a H-bond. Thr121 and Gly117 on helix IV are located at one or two helical turns away from the sugar-binding residue

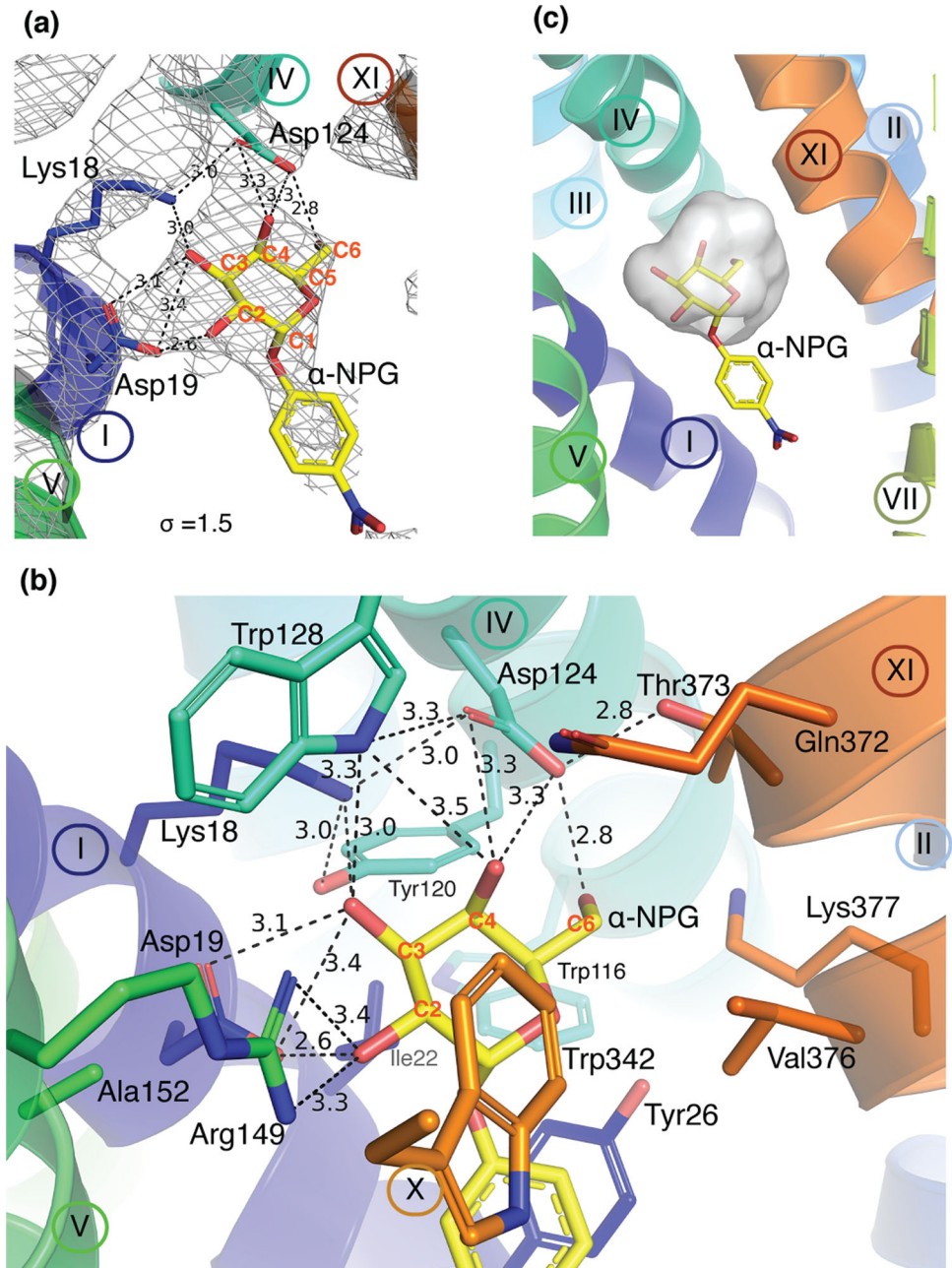

**Fig. 3 α-NPG-binding site.** The structure with bound α-NPG was refined to a resolution of 3.05 Å [PDB ID, 7L17]. Helices are shown in a cartoon representation and labeled in Roman numerals; protein sidechains are shown in sticks, labeled in three letters, and color-matched with hosting helix. This style is used throughout this article. Carbon positions on the galactosyl ring are labeled in red. H-bond and salt-bridge interactions are judged by distance that is ≤3.5 Å and indicated by dashed lines with distance shown in Å. The α-NPG molecule is shown in stick and the carbons are colored in yellow with oxygen in red and nitrogen in blue. **a** 2Fo-Fc electron-density map. A molecule of α-NPG is fitted to the electron density (contoured to 1.5 σ). Residues Asp19 and Lys18 (helix I) and Asp19 (Helix IV) form multiple strong H-bonds with all four OH groups on the galactosyl ring of α-NPG. **b** Sugar-binding site. Helix I (residues Lys18 and Asp124), helix IV (residues Tyr120, Asp124, Trp128), helix V (Arg149), and helix XI (residues Thr373) are involved in a charge-assisted H-bond network with all four OH groups on the specific galactosyl ring. Ile22 and Tyr26 (helix I), Trp342 (helix X), Val376 (helix XI) on helix IV are approximately 4 Å away and provide hydrophobic environment for galactosyl ring and/or phenyl ring. Few residues at longer distances are also shown up including Trp116 (helix IV) and Gln372 & Lys377 (helix XI). **c** Sugar specificity determinant pocket. The binding residues listed in panel **b** is shown in cavity presentation and colored in gray, where only hosts the galactosyl moiety of the α-NPG molecule.

Asp124. The OH on Thr121 and backbone O atom of Gly117 together with the two negatively charged Asp55 & Asp59 could form a negatively charged electrostatic surface on one side of the potential cation-binding cavity shaped by helices II, IV and XI, as well as C6-OH from the sugar. The H-bonded Thr373 and Asp124 located on the opposite surface of this cavity could potentially contribute to both sites simultaneously when MelB at

a different state, or contribute to Na$^+$ binding when the sugar is absent. Shifting both sites to a higher-affinity state could be achieved through conformational changes that bring the two sites closer allowing more interactions including a potential direct contact between sugar and cation.

As clearly demonstrated by the experimental results, an intact cation site should not be expected for this mutant. Nevertheless,

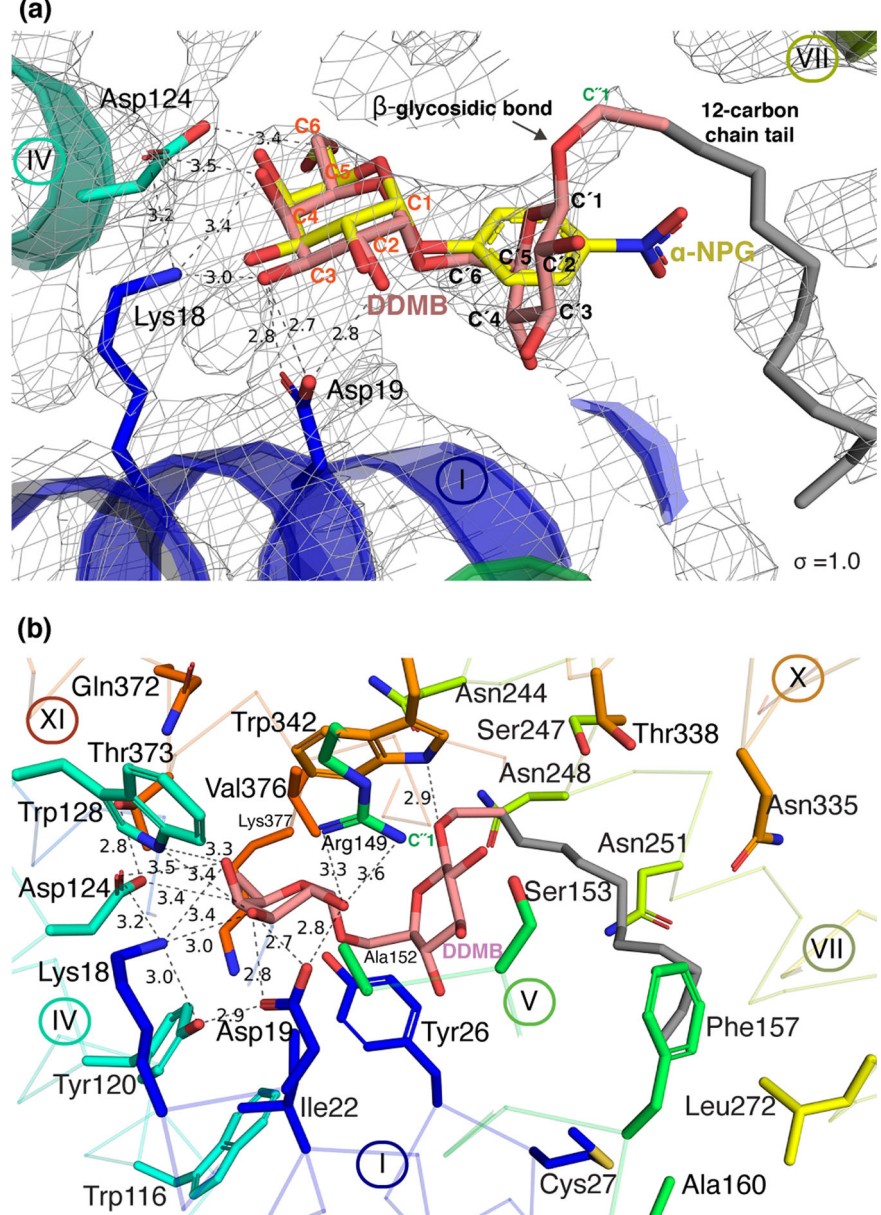

**Fig. 4 DDMB-binding site.** The structure with bound DDMB was refined to a resolution of 3.15 Å [PDB ID, 7L16]. Carbon positions are indicated (also presented in Fig. 1). The C3 to C12 on the detergent chain tail are presented in gray due to lack of electron density. H-bond and salt-bridge interactions are judged by distance ≤3.5 Å and indicated by dashed lines with distance shown in Å. **a** 2Fo-Fc electron-density map. One molecule of DDMB was fitted to the density map (contoured to 1.0 σ). The α-NPG in the α-NPG-bound structure, which was aligned to the DDMB-bound structure, is included for the comparison. **b** DDMB-binding site. Similar contacts between MelB_St and the galactosyl moiety observed in the α-NPG binding also exist in DDMB binding. Trp342 (helix X) forms a H-bond with C′1 O atom on the glucosyl ring; Lys18 adds one more H-bound to C4-OH of galactosyl moiety; Tyr120 adds one more bifurcated H-bond to Asp19; and C6-OH is at a longer distance to Asp124. Helices are shown in ribbon. Few more sidechains including Cys27 (Helix I), Ser153 and Ala 160 (helix V), and Asn244/248/251 (helix VII), and Asn335 and Thr338 (helix X) are at distances of less than 6 Å to the glucosyl moiety or the tail.

the cation binding pocket is contributed from three helices (II, IV, and XI) and both helical bundles, which is similar to the sugar specific pocket where five helices are involved (I, IV, V, X, XI). Furthermore, the intimate physical connection or potentially partial overlap between the two sites provides an essential structural basis for the established cooperative binding[24,45].

The bioinformatics analyses show that the galactoside-binding residues are only conserved within MelB orthologues from varied bacterial strains, and the positions involved in the cation site are highly conserved across nearly all members analyzed, including

those members for different substrates, such as MFSD2A and MFSD2B for phospholipids (Fig. 5c, d, Supplementary Fig. 6b). The human MFSD2A was modeled on the α-NPG-bound D59C MelB_St structure (7L17) based on a sequence identify of 18.47% (Fig. 5c). An outward-facing model was constructed by SWISS-MODEL covering positions 40 to 512, missing the first 39 residues and the last 18 residues at the C-terminal tail. In the proposed cation site (Fig. 5d), 4 out of 6 positions are identical except for Glu on the position Gly117 and Thr on Asn58 of MelB_St. These bioinformatics analyses seem to support the

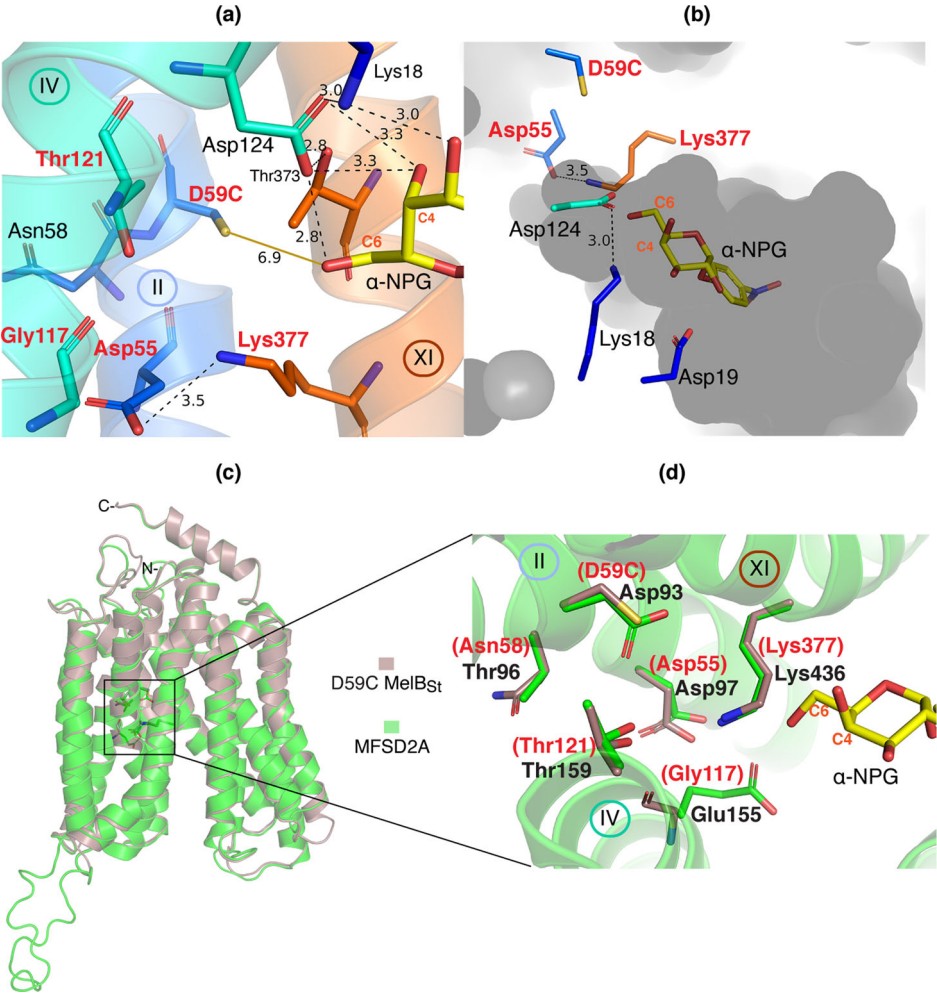

**Fig. 5 Proposed cation-binding pocket [PDB ID, 7L17].** H-bond and salt-bridge interactions are indicated by black dashed lines with distance shown in Å.
**a** Potential cation-binding site. Positions that have been determined to be involved in MelB cation binding including Asp55, Asp59, Gly117, Thr121, and Lys377 are labeled in red. Asp55 forms a salt-bridge with Lys377 at a distance of 3.5 Å. The C6-OH on the specific galactosyl ring is at a 6.9 Å distance to the sulfur atom on D59C, as indicated by the yellow line. Helices IV and XI carrying Tyr120, Asp124, Thr373, and Lys377 link the sugar specificity determinant pocket to the cation pocket. **b** A cross-section of surface representation of cation- and sugar- specific pockets. Internal solvent accessible cavity is shown in gray. Important charged residues (4 Asp and 2 Lys) and α-NPG molecule are shown in sticks. D59C is in a less solvent-accessible position than all other sidechains. **c** A 3-D model of the human MFSD2A. SWISS-MODEL server was used to model the outward-facing 3-D structure of the human MFSD2A (in green) using MelB D59C structure 7L17 as template (in dirty violet). **d** Potential cation site of the human MFSD2A overlaid with D59C MelB$_{St}$ mutant. The proposed cation site from the threading model in panel **c** was zoomed in to show the side chains (labeled in black text), which are conserved with the proposed cation site in MelB (labeled in red text).

observed Na$^+$ dependence in the Lyso-PC transport catalyzed by the Mfsd2a[3].

## Discussion

Results presented here and in literatures[9,47] have showed that D59C MelB mutant lost the cation binding and all three modes of cation-coupled melibiose transport, but retained the ability in galactoside binding and melibiose translocation[9,36,47]. This offered an opportunity to simplify the coupled symport to an uncoupled uniport. Crystallization of the WT MelB with bound sugar is quite challenging, probably due to the greater dynamic nature and more conformational states of being a symporter. With this uniport mutant carrying only sugar binding and translocation properties, the protein is more stable, crystallization trials are much more reproducible, and high-quality crystals can be obtained readily.

Overlay of the two structures with bound α-NPG or DDMB clearly reveals the sugar specificity determinant pocket is responsible for the galactoside recognition and binding in MelB (Figs. 2–4). The helices II (Lys18, Asp19, Ile22, and Tyr26, particularly Asp19 and Lys18) and VI (Tyr120, Asp124, and Trp128, particularly Asp124) make major contributions to the sugar binding, and helices V (Arg149), X (Trp342), and XI (Thr373 and Val376) also contribute to the binding affinity. Mutations on most of these positions greatly reduce the binding or transport in MelB, and some affected affinity to both sugar and cation[9,36,50–53]. The structure was also strongly supported by the Fourier-transform infrared spectroscopy (FTIR) studies in MelB$_{Ec}$, where the mutants on position-19 or 124 lost sugar-specific FTIR signals, and the D124C mutant also showed greatly reduced Na$^+$-specific signals[36].

There are only few disaccharide transporters with structure solved[55–57]. As shown in the sugar-binding site of LacY[55] and maltose-binding protein MalE in the maltose transporter

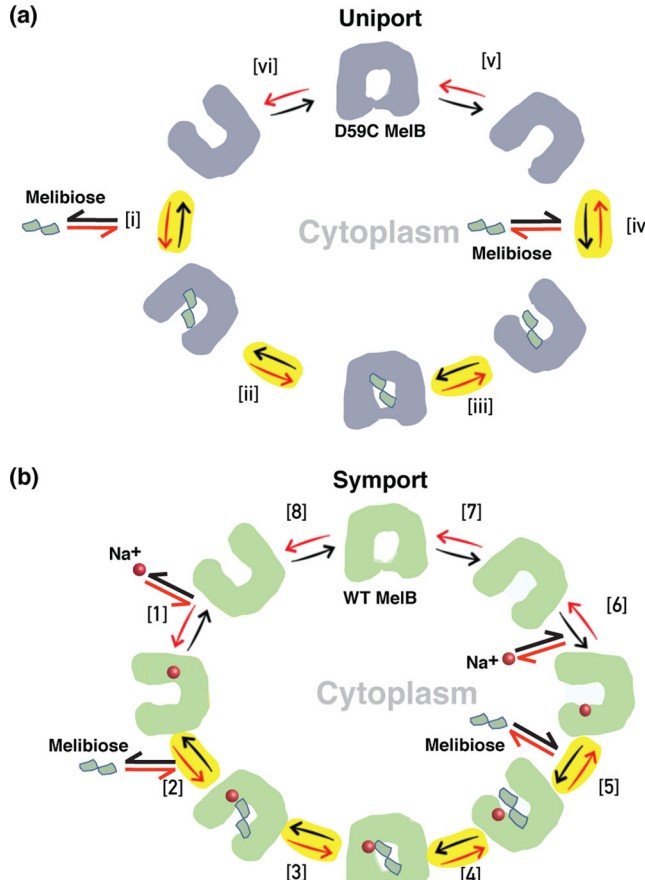

**Fig. 6 Transport models for the symport and uniport catalyzed by MelB.**
The symbols for the WT and D59C MelB proteins, Na$^+$, and melibiose are indicated. **a** Uniport catalyzed by the D59C MelB mutant. Melibiose inward-directly transport begins at step i and proceeds via the red arrows around the circle of 6 steps, with one melibiose across the membrane per circle. Melibiose efflux transport begins at step iv and proceeds via the black arrows around the circle, with one melibiose across the membrane per circle. Melibiose exchange begins at step iv, and takes one intracellular melibiose in exchange with one extracellular melibiose molecule, only involved in step iv – i, a 4-step reaction as highlighted in yellow color. **b** Symport catalyzed by WT MelB. Melibiose active transport begins at step 1 and proceeds via the red arrows around the circle, with one melibiose and one cation inwardly across the membrane per circle. Melibiose efflux transport begins at step 6 and proceeds via the black arrows around the circle, with one melibiose and one cation outwardly across the membrane per circle. Melibiose exchange begins at step 6, and also takes 4 steps involved step 5 - 2 as highlighted in yellow color.

MalEFGK$_2$[58], charged residues dictate the sugar binding, but it is not the case for the low-affinity maltose binding site within the membrane-domain in MalF[57]. Interestingly, an "acceptor-donor-acceptor" pattern for H-bonding with sugar is also found in MalE[58]. Aromatic residues presented in these sugar-binding sites form CH/π-interactions between aromatic and the galactosyl rings, or aromatic stacking interactions with phenyl ring, which shape the binding pocket and increase the interaction strength[59,60]. Different from LacY, MelB sugar-binding site displays a comprehensive salt-bridge-assisted H-bonding network; the sugar arrangement in the two respective sugar-binding sites is also different. While both permeases transport galactosides, the sugar OH position connecting to its coupling cation site differs. In LacY, the C3 and C2-OH groups link to the H$^+$ site[55,56], while the C6-OH is in close proximity to MelB cation site (Fig. 5).

The two structures can explain well why MelB exhibits a broader sugar specificity, and recognizes a large variety of galactosides from mono- to tri-saccharides, as well as other glycoside analogs (methyl-thiogalactosides, methyl-galactosides, NPG, dansyl-galactoside with 2 to 6 carbons between the galactosidic and the dansyl moieties)[26,28,35]. A galactosyl moiety, which is common for all galactoside sugars, confers the specificity to be recognized by the sugar specificity pocket, whereas the structural/chemical variation on the non-galactosyl moiety can be accommodated by the large non-specific binding cavity. The non-specific interactions can largely increase the binding affinity. This structural insights into the galactoside binding might suggest a potential drug delivery strategy based on glycosides being transported by a sugar active transporter.

Determination of the sugar-binding site in MelB allows to recognize the nearby cation specificity determinant pocket (Fig. 5a, b, Supplementary Fig. 7a), which confirms the previous conclusion based on a large body of biochemical/biophysical analyses including mutagenesis and second-site suppressors, all favoring the overlapping sites for sugars and cations[36,50,51,53,61,62]. While the Na$^+$ binding was not resolved with MelB, as mentioned above, several positions including 55, 59, 117 and 121 have been determined essential or important for the binding of Na$^+$ and Li$^+$, as well as for the coupled transport[9,24,36–38,46,47,63,64]. In addition, Asn58 at the neighboring position of 59 play important role in the cation selectivity because Ala58 in *Klebsiella pneumonia* MelB selectively eliminate the Na$^+$ recognition whereas retains the H$^+$- and Li$^+$-coupled transport[65,66]. It is also noteworthy that Arg replacement on Gly117 decreases affinity for Na$^+$ and Li$^+$ by about 10-fold, fails in melibiose active transport and efflux, but catalyzes melibiose exchange coupled to Na$^+$ or Li$^+$ at a normal rate[38]. Cys replacement on Thr121 selectively inhibit Na$^+$-coupled melibiose transport with less effect on Li$^+$-coupled melibiose transport[9]. More functional studies are still needed to assign the specific roles of these positions in the cation site; ultimately, a crystal structure showing the Na$^+$ binding will reveal more details of this interesting cation site in MelB.

In the human MFSD2A, the substitution of Met for Thr at the conserved position 121 in MelB resulted in a lethal mutation[67] with no Na$^+$-dependent transport. It is likely that the mutation affected Na$^+$ binding (Fig. 5c, d). In addition, S166L mutation in MFSD2A is another lethal mutation[67] and the equivalent position Trp128 in MelB$_{St}$ plays important role to stabilize the sugar binding by forming H-bond interactions with C2- and C3-OH groups, as well as with the substrate-binding critical side chain on Asp124. It is possible that Ser166 of MFSD2A and its spatially neighboring residues may form a phospholipid specificity determinant pocket in close proximity to the cation site, and Ser166 may play an important role for the phospholipid recognition. For another human homolog MFSD13A, four possible cation-site positions, as underlined, are conserved as shown in MelB sequence ($^{63}$NSL<u>N</u><u>D</u>$^{67}$, $^{135}$DGFL<u>T</u>LV<u>D</u>$^{124}$, and <u>K</u>$^{412}$) (Supplementary Fig. 7b). It is likely that those human homologues are cation-coupled symporters[3], while the substrate and coupling cation species are still unknown for MFSD13A.

In MelB, all three cations (H$^+$, Na$^+$, or Li$^+$) compete for the same site with a stoichiometry of unity. While a metal site could be conceptualized based on the structure, where is the site for H$^+$? Competitive Na$^+$ binding at varied pH revealed p$K$a values of 6.25 or 6.59 for MelB$_{St}$ in the absence or presence of melibiose[24], and p$K$a value of 6.3 for MelB$_{Ec}$[68]. Asp55 is salt-bridged with Lys377, and both are solvent accessible (Fig. 5b). The p$K$a for Asp55 is expected to be at a normal range, which could argue against for serving as the protonation site. Asp59 is located in a less-solvent exposed position (Fig. 5b) so an elevated p$K$a value for this carboxyl group is expected. More direct evidence could be

the result that D59C MelB$_{Ec}$ lacks transient alkalization of medium during melibiose gradient-driven melibiose transport[47]. Thus, Asp59 is likely the H$^+$-binding residue in MelB; thus, the D59C mutant should lack the affinity for all three cations.

The position 59 regulates MelB between symport and uniport for melibiose transport; a transport kinetic model for both can be simplified as an 8-step or 6-step cycle, respectively (Fig. 6a, b). For the uniport process (Fig. 6a), melibiose is recognized [i] by the apo mutant, which induces alternating-access process and shifts MelB conformation into an occluded intermediate [ii] and opens to the cytoplasm [iii]. Bound melibiose is released into the cytoplasm [iv], and MelB closes the cytoplasmic opening [v] and resets to the outward-open state [vi] followed by a cycle. Since the $K_d$ value for melibiose binding to D59C mutant is as low as 6 mM, it can only work when environmental melibiose at high concentrations, such as 30 mM used in the MacConkey agar or in melibiose exchange experiment[9]. Under active transport assay, melibiose was presented at 0.4 mM, so no uptake is detected (Fig. 1). At step [iv], melibiose release or rebound is dependent on the intracellular melibiose concentration. In the fermentation assay, the incoming melibiose is immediately hydrolyzed by a high-turnover enzyme α-galactosidase, so melibiose concentration can not be built up and melibiose cytoplasmic release should be largely favored over the rebound. For the symporter process, two more steps are added including cation binding and release (Fig. 6b), which allow MelB to work at environments with low-level melibiose concentrations. This is advanced by increasing the melibiose binding affinity through a positive cooperativity mechanism[24]. In the presence of an electrochemical ion gradientce of H$^+$, Na$^+$, or Li$^+$, the cytoplasmic release of bound cation at step [6] is greatly favored, which should facilitate the cytoplasmic closure and prevent melibiose rebound, so the transport $K_m$ can be largely improved[29,37,69].

In summary, this structural study shows that the specificity determinants of galactosides and cations in MelB are very close and contributed from both helical bundles. The close connection between the two specificity determinant pockets laid the structural foundation for the obligatory coupling of this transporter family. The structural findings on MelB characteristic sugar recognition mechanism, using a narrow specific binding groove and a large non-specific pocket, could offer a potential for future exploration of active transporters for drug design and delivery.

## Materials and methods

**Materials and reagents**. [1−$^3$H]Melibiose was custom-synthesized by Perki-nElmer (Boston, MA). Other chemicals and reagents used in this study were of analytical grade and purchased from standard commercial sources except 2′-(N-dansyl)aminoalkyl-1-thio-β-D-galactopyranoside (D$^2$G) that was kindly provided by Drs. Gerard Leblanc and H. Ronald Kaback. The detergents undecyl-β-D-maltopyranoside (UDM), dodecyl-β-D-maltopyranoside (DDM), and dodecyl-β-D-melibioside (DDMB), and E. coli polar lipids (Extract Polar, Avanti,100600) were purchased from Anatrace. Melibiose and 4-nitrophenyl-α-D-galactopyranoside (α-NPG) were purchased from Sigma-Aldrich. The SelenoMet Media were purchased from Molecular Dimensions Limited. Crystallization reagents and materials were purchased from Hampton Research.

**Plasmids and cell culture for transport assays**. The overexpression of MelB$_{St}$ was carried out in the E. coli DW2 strain (melA$^+$, melB$^-$, and lacZ$^-$Y$^-$) from a constitutive expression plasmid pK95ΔAH/MelB$_{St}$/CHis$_{10}$[9,28,33]. The D59C or D59A mutant with Cys or Ala at the position 59, respectively, was constructed previously[9] or in this study by QuikChange Site-Directed Mutagenesis kit and confirmed by DNA sequencing analysis. E. coli DW2 cells containing a given plasmid were grown in Luria-Bertani (LB) broth (5 g yeast extract and 10 g tryptone per liter with 171 mM NaCl) with 100 mg/L of ampicillin in a 37 °C shaker. The overnight cultures were diluted by 5% with LB broth supplemented with 0.5% glycerol and 100 mg/L of ampicillin, and constitutive overexpression was obtained by shaking at 30 °C for another 5 h.

**Melibiose fermentation**. The DW2 cells were transformed with a given plasmid, plated on MacConkey agar plate supplemented with melibiose at 30 mM (the sole carbohydrate source) and 100 mg/L ampicillin, and incubated at 37 °C. After 18 h, the plates were viewed and photographed immediately.

**[1-$^3$H]Melibiose transport assay**. E. coli DW2 cells expressing MelB$_{St}$ were washed with 100 mM KP$_i$, pH 7.5, to remove Na$^+$ contamination as described previously[28]. The cell pellets were resuspended with the assay buffer (100 mM KP$_i$, pH 7.5, 10 mM MgSO$_4$) and adjusted to $A_{420}$ of 10 (~0.7 mg protein/ml). Using a fast filtration, transport time course at 0.4 mM melibiose (at specific activity of 10 mCi/mmol) was carried out in the absence or presence of NaCl or LiCl at 20 mM or 100 mM, respectively, and intracellular melibiose was determined as described previously[28].

**Large-scale protein production**. LB broth supplemented with 50 mM KPi (pH 7.0), 45 mM (NH$_4$)SO$_4$, 0.5% glycerol, and 100 mg/L ampicillin was used for the large-scale expression in fermenters, and protocols for membrane preparation and MelB$_{St}$ purification by cobalt-affinity chromatography after extracted in detergent UDM have been described previously[9]. MelB$_{St}$ protein solutions were dialyzed overnight against a buffer (consisting of 20 mM Tris-HCl, pH 7.5, 100 mM NaCl, 0.035% UDM, and 10% glycerol), concentrated with Vivaspin column at 50 kDa cutoff to approximately 40 mg/mL, and subjected to ultracentrifugation at 384,492 g for 45 min at 4 °C (Beckman Coulter Optima MAX, TLA-100 rotor), stored at −80 °C after flash-frozen with liquid nitrogen.

**Seleno-methionine incorporation**. The seleno-methionine derivative of MelB$_{St}$ D59C mutant (SeMet D59C MelB$_{St}$) was obtained by incorporating Sel-Met during the protein expression in the E. coli strain DW2 in the SelenoMet Medium (Molecular dimensions). When supplemented Met at a ratio of 10:1 for Sel-Met vs. Met to the medium, the SeMet D59C MelB$_{St}$ sample failed to produce crystals, so partial labeling at a ratio of 3:1 (66.7% sel-Met) was applied.

**Protein concentration assay**. The Micro BCA Protein Assay (Pierce Bio-technology, Inc.) was used for the protein concentration assay. Protein concentration for crystallization trails were estimated by measuring the UV absorption at 280 nm.

**CD spectroscopy**. MelB$_{St}$ at 10 μM in 10 mM NaPi, pH 7.5, 100 mM NaCl, 10% glycerol, and 0.035% UDM was analyzed with Jasco J-815 spectrometer equipped with a peltier MPTC-490S temperature-controlled cell holder unit. The CD spectra for a wavelength range of 200–260 nm were carried as described previously[49]. Melting temperature ($T_m$) determination was carried out at temperatures between 25–80 °C. The CD spectra and ellipticity at 210 nm were recorded at an interval of 2–5 °C or 1 °C, respectively, with the temperature ramp rate at 1 °C per minute, and plotted against the temperature. The ellipticity at 210 nm is decreased along with temperature increase reflecting helical unwinding, partial unwinding, and/or protein precipitations out of the solutions. The $T_m$ values were determined as the temperature leading to the half maximal increase in the unfold fractions including precipitations by fitting the data using the Jasco Thermal Denaturation Multi Analysis Module.

**Isothermal titration calorimetry (ITC) measurements**. All ITC ligand-binding assays were performed with the TA Instruments (Nano-ITC device) at 25 °C. In a typical experiment, the titrand (MelB$_{St}$) in the ITC sample cell was titrated with the specified titrant (placed in the Syringe) by an incremental injection of 2-μL aliquots at an interval of 300 s at a constant stirring rate of 250 r.p.m. All samples were degassed using a TA Instruments Degassing Station (model 6326) for 15 min prior to the titration. Data processing was performed with the NanoAnalyze (version 3.6.0 software) provided with the ITC equipment. The normalized heat changes were subtracted from the heat of dilution elicited by last few injections, where no further binding occurred; the corrected heat changes were plotted against the molar ratio of titrant versus titrand. The values for association constant ($K_a$) value were determined by fitting the data with a one-site independent-binding model. At most cases, the binding stoichiometry (N) number was fixed to 1 since it is a known parameter, which can restrain the data fitting and achieve more accurate results[70]. The dissociation constant $K_d = 1/K_a$.

**Trp→dansyl fluorescence resonance energy transfer (FRET) and IC50 determination**. Steady-state fluorescence measurements were performed with an AMINCO-Bowman series 2 spectrometer with purified MelB$_{St}$ at 1 μM in 20 mM Tris-HCl, pH 7.5, 100 mM CholCl, 10% glycerol, 0.03% UDM. Using an excitation wavelength at 290 nm, the emission intensity was recorded at 490 nm. On a time-trace, D$^2$G, NaCl or LiCl, then DDMB or α-NPG, were sequentially added into the protein solutions at a 1 min-interval. For the determination of IC$_{50}$, after the addition of 10 μM D$^2$G (the $K_d$ for the WT), titrants (DDMB, DDM, UDM, or α-NPG) was consecutively added until no change in fluorescence intensity occurred. An identical volume of water was used for the control. The decrease in intensity after each addition of DDMB or α-NPG was corrected by the dilution effect and plotted as a function of the accumulated α-NPG concentrations or estimated DDMB concentrations. The 50% inhibitory concentration (IC$_{50}$) was determined

**Table 3 Crystallographic Data collection, phase, and refinement statistics.**

| Data collection | SelMet D59C MelB$_{St}$ with DDMB | D59C MelB$_{St}$ with α-NPG [PDB ID, 7L17] | D59C MelB$_{St}$ with DDMB [PDB ID, 7L16] |
|---|---|---|---|
| Space group | P 31 2 1 | P 31 2 1 | P 31 2 1 |
| Cell dimensions | | | |
| a, b, c (Å) | 126.4 126.4 103.7 | 126.364 126.364 104.334 | 127.372 127.372 103.905 |
| α, β, γ (°) | 90 90 120 | 90 90 120 | 90 90 120 |
| Resolution (Å) | 38.4–4.0$^a$ | 29.89–3.05$^b$ | 29.35–3.15$^b$ |
| $R_{sym}$ or $R_{merge}$ | 0.185 (1.213) | 0.111 (1.404) | 0.138 (1.463) |
| $I / \sigma I$ | 77.4 | 20.5 (2.4) | 18.2 (2.6) |
| Completeness (%) | 99.41 | 99.48 (99.95) | 98.66 (100.00) |
| Redundancy | 64.4 (50.51) | 21.6 (22.5) | |
| **Refinement** | | | |
| Resolution (Å) | | 29.35–3.05 (3.159–3.05) | 29.35–3.15 (3.263–3.15) |
| No. reflections | | 18614 (1862) | 16973 (1684) |
| $R_{work} / R_{free}$ | | 0.254 / 0.287 | 0.273 / 0.297 |
| No. atoms | | 3582 | 3596 |
| Protein | | 3561 | 3561 |
| Ligand/ion | | 21 | 35 |
| Water | | 0 | 0 |
| B-factors | | | |
| Protein | | 101.91 | 103.07 |
| Ligand/ion | | 101.94 | 102.92 |
| | | 96.56 | 117.89 |
| R.m.s. deviations | | | |
| Bond lengths (Å) | | 0.003 | 0.002 |
| Bond angles (°) | | 0.615 | 0.520 |

$^a$Merged from 13 datasets.
$^b$A single crystal was used for the structure. Values in parentheses are for highest-resolution shell.

by fitting a hyperbolic function to the data (OriginPro 2020b). The final concentration of each detergent is estimated to be 234 μM, which is less than their CMC value for (DDMB, 295 μM). It is noteworthy that the actual concentration for monomeric form under this experimental setup with protein in UDM solution is unknown, so the concentration should be overestimated and yield an underestimated binding affinity.

**Crystallization, native diffraction data collection, and processing**. Crystallization trials were carried out by the hanging-drop vapor-diffusion method at 23 °C by mixing 1 μL of the pre-treated protein samples containing α-NPG or DDMB with 1 μL reservoir. To prepare the α-NPG-containing D59C MelB$_{St}$, the protein samples in the dialysis buffer composed of 20 mM Tris-HCl, pH 7.5, 100 mM NaCl, 0.035% UDM, and 10% glycerol was diluted to 10 mg/mL and supplemented with phospholipids at a concentration of 3.6 mM (E. coli Extract Polar, Avanti, 100600) from a 20-mM stock dissolved with a dialysis buffer containing 0.01% DDM instead of 0.035% UDM, as well as 6 mM α-NPG. To prepare DDMB-containing D59C MelB$_{St}$, the same treatment was carried out except that the protein samples were supplemented with 0.015% DDMB (1 x CMC) and 10% PEG3350. All samples were incubated for 15 min prior to the crystallization trials, and both crystals of D59C MelB$_{St}$ containing α-NPG or DDMB were grown using a reservoir consisting of 50 mM BaCl$_2$, 50 mM CaCl$_2$, 100 mM Tris-HCl, pH 8.5, and 29–32% PEG 400. The crystals can be obtained from a wide range of reservoir conditions with no notable change on the structure; such as, 50 mM BaCl$_2$ can be replaced by 100 mM NaCl, or 100 mM Tris-HCl, pH 8.5 can be replaced by 100 mM MES, pH 6.5. Crystals appeared in 3–4 days, were frozen in 2–3 weeks with liquid nitrogen, and tested for X-ray diffraction at the Lawrence Berkeley National Laboratory ALS BL 5.0.1 or 5.0.2 via remote data collection method. The complete diffraction datasets for α-NPG- or DDMB-bound native crystals were collected at 100 K from a single cryo-cooled crystal at a wavelength of 0.97949 Å on ALS BL 5.0.2 or 0.97741 Å on ALS BL 5.0.1, respectively, with a Pilatus3 6 M 25 Hz detector. ALS auto-processing XDS output files were used for structure solution. The statistics in data collection is described in Table 3.

**Anomalous diffraction data collection, processing and experimental phasing**. The crystals from SeMet D59C MelB$_{St}$ mutant were obtained from 50 mM BaCl$_2$ or 100 mM NaCl, 50 mM CaCl$_2$, 100 mM Tris-HCl, pH 8.5, or 100 mM MES, pH 6.5. Weak anomalous signals presented in each individual dataset, and multiple datasets at varied wavelength were collected and processed either manually by HKL2000[71] or DIALS in CCP4i2[72], or using the ALS auto-processing DIALS method. Phenix Scale and Merge Data and Phenix Xtriage programs (v.1.18.2-3874 or dev-3936)[73] were used for selecting and merging datasets. A merged dataset, which was generated from 13 datasets collected from 6 crystals with a resolution cutoff between 40–4.0 Å, exhibits anomalous signal with CC$_{ano\ half}$ >0.94 at 8.0 Å and 0.04 at 4 Å, as analyzed by Phenix Anomalous Signal program. Among those data sets, most

were from a suboptimal wavelength for Se atom (0.977408 Å) collected on BL5.0.1, and one from a wavelength of 0.97919 Å on BL5.0.2.

Searching for selenium atoms and phasing at a several resolutions limited to 6.0 to 6.5 Å (CC$_{ano\ half}$ >0.44) were performed using ccp4i2 (7.0) Crank2 Phasing and Building program[74] and Phenix Hybrid Substructure Search program[75,76]. A total of 18 selenium sites were identified, and used for phasing and model auto-building by SAD or MR-SAD using a model derived from the pdb id 4M64 by the Crank2 Phasing and Building programs. The model building was performed in Coot 0.8 or 0.9[77] and refined against the combined anomalous dataset at 4.0 Å by Phenix Refinement[78].

**Structure determination of native D59C MelB$_{St}$ and α-NPG or DDMB fitting**. The structure determination for the native data sets was performed by molecular replacement used the refined SeMet model as the search model in the Phenix Phaser-MR[75], and followed by rounds of manual building and refinement. Overlay of the 18 sites of selenium atoms is shown in the Supplementary Fig. 4b. Ligand fit using α-NPG 3D structure with a ligand code 9PG from pdb id 4zyr[55] was performed with Phenix LigandFit program[79] for the well-refined models. The 3-D structure of DDMB was created by a 2D-structure drawn by Chemdraw; Phenix eBLOW[80] and Readyset were used to generate and optimize the ligand restraints for refinement with a code of LMO. The manually aligned DDMB structure on α-NPG was used for LigandFit[81]. All ligand-bound structures were modeled from positions 2 to 454 without a gap. After rounds of refinement, the structures with α-NPG or DDMB were refined to a resolution of 3.05 Å (PDB access ID, 7L17) or 3.15 Å [PDB access ID, 7L16], respectively. Several crystal structures of D59C mutant bound with α-NPG or DDMB were refined to resolutions beyond 3.3 Å, and all structures are virtually identical. The electron density for each ligand was analyzed by unbiased composition omit map and Polder map[81] (Supplementary Fig. S6a–d). Met at the first position was not present due likely to the post-translational processing, which has been verified by N-terminal sequencing analyses with MelB$_{St}$ protein samples used for the structural determination (Alphalyse Inc. CA). The statistics of refinement for the final models were summarized in Table 3. For α-NPG- and DDMB-bound structures: Ramachandran favored of 98.23% and 95.57%, Ramachandran outliers of 0%, clash scores of 1.38 and 2.19, and overall scores of 0.87 and 1.30, respectively, as judged by MolProbity in Phenix. Pymol (2.3.5) was used to generate all graphs.

**Modeling of the human MFSD2A**. A 3-D structure of human MFSD2A was modeled by the homology-modeling server SWISS-MODEL using the D59C MelB$_{St}$ structure (7L17) as an input template.

**Statistics and reproducibility**. All experiments were performed 2–4 times. The average values were presented in the tables with standard errors.

**Reporting summary**. Further information on research design is available in the Nature Research Reporting Summary linked to this article.

## Data availability

The protein coordinates and structure factors have been deposited into the Protein Data Bank with the accession numbers 7L16 and 7L17 for the D59C MelB$_{St}$ bound with DDMB or α-NPG, respectively, which will be released immediately upon the publication of this manuscript. The raw data for all figures and tables are either available in Supplementary Data file or from the corresponding author on reasonable request.

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

## Acknowledgements

The authors thank Drs. R. Bryan Sutton, Luis Cuello, Thomas Terwilliger, Abdul Ethayathulla, and Kei Nanatani for discussion and help. The strain DW2, the expression vector, and reagent D$^2$G were kindly obtained from Drs. Gerard Leblanc and Ronald Kaback. This study is supported by the National Institutes of Health grants R01GM122759 and R21NS105863 to L.G.

## Author contributions

L.G. designed this study and performed protein crystallization, X-ray diffraction data collection and processing, structure determination, interpretation, and figure preparations. P.H. performed all protein purification, functional characterizations and related data processing, and figure preparations. L.G. wrote the manuscript with help of P.H.

## Competing interests

The authors declare no competing interests.
