## [Transparent Peer Review File · Communications Biology]

Reviewers' comments:

Reviewer #1 (Remarks to the Author):

In this study, Lan Guan and the co-author report two crystal structures of melibiose transporter MelB in complex with substrate analogs, nitrophenyl-galactoside, and dodecyl-melibioside. They crystallized a MelB mutant D59C, which helps retain ligands in the binding pocket. The work is interesting and it may help to understand the substrate binding mechanism of MelB. However, there are several concerns regarding this work:

Major concerns:

1. In Fig. 1a, the authors did not detect any cation-dependent transport activity with D59C and they claim this mutant is a uniporter. Can they confirm it using the 3H-melibiose transport assay? Is it possible that D59C is inhibited by cations or just an inactive mutant?
2. The authors need to provide composite omit maps to confirm the presence of each ligand in the structures, which are important to avoid any model bias. Only 2Fo-fc maps are provided in Fig. 3 and 4.
3. The authors describe the ligand-binding mechanism in great atomic details on pg. 8-10. They need to be careful given the modest resolution (3 angstroms) of the structures. However, no functional data support it. Can they confirm the binding mechanism by mutations?
4. It is a bit surprising that no structure comparison is provided between the apo and ligand-bound forms. Any conformational changes between these two conformations? Any conformational changes induced by ligand binding in the substrate-binding pocket? This information is very important to understand the transport mechanism.
5. What is the purpose to use both nitrophenyl-galactoside and dodecyl-melibioside? Both structures are nearly identical and have similar binding conformation. No much more information from these duplicate works.

Minor concerns:

1. In the abstract, the authors state that "the conserved cation-binding pocket is also assigned". How to assign the cation-binding pocket if they did not see any cation in the structure giving the fact that the D59C mutation has abolished cation binding?
2. The manuscript needs substantial revision. The authors should focus on their own work instead of discussing too much about other human transporters. For example, the first paragraph can be summarized in a couple of sentences. I understand that MelB belongs to the MFS₂ family, which contains several human transporters. However, lipid transporter MFSD2A may have a completely different mechanism. The authors need to be careful to avoid overstatement. Also need to improve the English. Many typos and grammatical errors make it hard to follow.

Reviewer #2 (Remarks to the Author):

Guan and Hariharan present the first high resolution structures for ligand bound sodium-dependent MFS symporter, melB. The rationale for this work is to determine in greater detail the transport mechanism for this important class of secondary transporters, which has been lacking. This findings from this study are important across multiple fields of research because MFS proteins play critical roles

in physiology in multiple kingdoms of life and are particularly important in human diseases and delivery of therapeutics to cells and tissues. MelB in particular has served as one of the model MFS proteins with which to understand transport mechanisms for this family. In this report, the authors extend their long-time work on MelB utilising both highly quantitative methods for measuring transport and conformational changes (e.g. binding and competition studies, ITC analysis) and crystallography. Two different sugar ligands α -NPG and DDMB were captured in two high resolution structures of a transport dead mutant (D59C at 3 and 3.15 Angstroms, respectively). Both structures revealed similar sugar coordinating or binding residues and resolved the cation binding pocket. Although sodium was not captured in these structures because D59C was used, this study revealed the close interaction between the sugar binding and cation binding sites.

Overall, the data analysis and biochemical studies support the main conclusions of this paper. This work is timely and important to multiple fields of research that span interests in transporters and medicine and can be used as a roadmap to better understand how other clinically relevant MFS transporters work. In this regard, this work could be improved by addressing the following comments/suggestions.

- 1) Does the fact that the D95C structure capture sugar ligand binding indicate that it is possible that sugar binding can precede cation binding in the transport cycle?
- 2) DDMB exhibited 4X higher affinity than α -NPG in the presence of sodium. The authors reveal a "non-specific" hydrophobic pocket. Does this pocket allow for increased binding affinity for DDMB. The reason for this question is that the authors mention Mfsd2a which is a lysolipid transporter and based on modelling against MelB has a much larger hydrophobic pocket than MelB. Mfsd2a can also transport LPCs with 12 carbon acyl chains (as DDMB contains). Is this pocket in MelB and Mfsd2a in similar regions of the protein and composed of similar transmembrane domains?
- 3) Does the PDB 4m64 structure and the other structure (partially occluded from the Guan lab, Nat Comm paper) in comparison to these new structures (apo vs holo) reveal any subtle conformational changes in the ligand binding pocket?
- 4) Is the S166 residue of human Mfsd2a which is mutated to S166L in familial autosomal microcephaly 15 homologous to S153 in MelB? The reason for this question is that there is little understanding of the role of S166 in Mfsd2a, and it might be possible to answer this question with these new structures.
- 5) Can the authors comment on how they predict the cation transport path might look like for MelB and why Mfsd2a is strictly sodium dependent while MelB is not. Again, these questions are intended to improve the impact of this work and can be addressed with modelling or discussion.
- 6) In the results section on ligand binding, can the authors comment on the functional testing of the residues that have been identified to coordinate sugar binding? I believe many of these residues have been tested for function in the authors' previous studies and would be helpful to readers not familiar with MelB to point this out.
- 7) It would be more appealing to the broader transporter community to modify the final sentence in the abstract as "These key structural findings lay a strong basis for understanding cooperative binding and symport mechanisms for sodium-dependent MFS transporters, including eukaryotic transporters such as Mfsd2a."

Reviewer #3 (Remarks to the Author):

Guan and Hariharan present x-ray crystallographic structures of the membrane protein MelB. MelB is a secondary transporter from the Major Facilitator Superfamily (MFS), and normally transport the dicccharid melibiose couple to the symport of Na^+ or H^+ (the non-physiological Li^+ can also drive the active transport of melibiose). The capability of using both Na^+ and H^+ to drive the transport makes MelB unique among the MFS. MelB has become also recently relevant due to its homology to some human secondary transporters: MFSD2A and MFSD13a. MelB was extensively studied, mainly in

the 80s and 90s, arguably being, after the Lactose permease, the best-understood secondary transporter. However, it resisted crystallization, and its two first structures with atomic resolution were non published until 2014. These two structures lacked both the sugar and the cation, although the residues involved in the binding site could be indirectly assigned thanks to the large amount of previous work.

In this occasion, Guan and Hariharan present two X-ray structures of a MelB mutant (D59C) bound to two high-affinity melibiose analogues: NPG and the detergent DDMB. Although this mutant lacks both Na⁺ and H⁺ binding, it shows for the first time how the galactoside ring of melibiose binds to the protein, in what the author call the galactoside pocket. It also shows a larger nonspecific binding pocket, which explains why MelB binds not only disaccharides but also trisaccharides or detergent molecules as DDMB. The results fits well with what was known about which are the important residues for melibiose binding, but with a level of detail that only structural data at atomic resolution can achieve. Therefore, it is a important step further in understanding MelB and related transporters. The paper is well written and well-illustrated by the included figures.

In spite of my general positive impression, I have some questions/comments about aspects that I believe that require some clarifications, and perhaps some corrections.

1. In the Results, the author switch from [3H]melibiose transport assays to ITC and fluorescence measurements without any clarification about the state of the protein. For the [3H]melibiose transport assays it would be better to specify that those are performed in whole-cells (to my understanding), while the rest of experiments (ITC, FRET, CD) are conducted in detergent solubilized MelB. This is an important clarification, as being a membrane protein without any further explanation readers might jump into the assumption that all experiments are done for MelB in lipidic membranes.

2. The titration experiments of DDMB, DDM and UDM using FRET casts several questions. Regarding DDMB it is said that its concentration is below the cmc, to keep it monomeric. However, the cmc is defined when the only component in a solution is water, and will be affected by the presence of other detergents. It is very likely that the DDMB, even if below the cmc, will be incorporated in the UDM micelles that solubilize MelB. Another issue is the use of DDM and UDM as a negative control, showing that the binding of DDMB is driven by its melibiosyl moiety and did not occur when changed to a maltosyl. But the authors do not explain how these experiments were conducted. If DDM binding was tested for MelB solubilized in UDM, it is reasonable not to detect any binding, as any potential binding sites would be occupied by the UDM in excess. I can only guess that when testing DDM and UDM binding MelB was solubilized in a detergent lacking a sugar moiety.

3. The experiments testing the thermoestability fo MelB WT and D59C are disturbing. They show how the ellipticity at 220 nm, related to the helicity, drops to almost zero above a certain temperature, indication that MelB loses its helical structure at high temperature. This is nonsense, as helical membrane proteins remain helical even when detaurated at high temperature. The "unfolding" of membrane proteins usually involves aggregation of loops, and rearrangement in the transmembrane helices, but without or very small changes in helicity. My only guest is that the CD signal goes to zero because the protein precipitates. In any case, the authors should include the CD spectrum at temperature above the T_m, and UV-vis or fluorescence spectra below and above the T_m, to confirm that the protein remains in solution.

4. Somewhere in the MS appears "The373". Thr373?

5. In the "proposed cation-binding pocket" the author, talking about Lys377, they say: "It could switch between a salt-bridge with Asp55 and a H-bond with C6-OH on the sugar." This possibility is not supported by their data, but has been shown to be possible in MD simulations (Fuerst et al (2015) JBC). To my knowledge, the above work is the only one displaying how the cation-binding pocket of MelB could look like, so it should be taken into account in this section.

6. At the end of the Results the authors mention the MFSD2A and MFSD2B transporters, giving a reference to Fig. S5, but the correct figure is Fig. S6.

7. All the characterization of the binding properties of MelB are done in detergent solubilized samples in the presence of 10% glycerol. Why glycerol was added? Why so much? MelB is a stable protein even in detergent, so I cannot believe it was necessary to add glycerol to make the ITC or FRET possible. Is there any other reason?

8. In the FRET experiments MeIB is not only in the presence of 10% glycerol but also in the presence of 100 mM CholCl. What is CholCl? Why was it added?
9. At the end of the "Methods" there is a sentence I cannot understand: "The first met was not present as also shown by N-terminal sequencing analyses with MeIBSt (Alphalyse Inc. CA)."

Point-by-point rebuttal letter

Reviewer #1 (Remarks to the Author):

Major concerns:

1. In Fig. 1a, the authors did not detect any cation-dependent transport activity with D59C and they claim this mutant is a uniporter. Can they confirm it using the 3H-melibiose transport assay? Is it possible that D59C is inhibited by cations or just an inactive mutant?

The [³H]melibiose transport with intact cells in the absence or presence of Na⁺ or Li⁺ at two concentrations were presented in the original Figure 1a. The transport activity of the mutant as judged by fermentation (reflecting rate of melibiose uptake) was also presented in Figure 1. Per the editor's requested, we specifically constructed a D59A mutant for this revision. Both D59A and D59C mutants are melibiose uniporter.

The D59C MelB_{St} mutant loses the abilities to bind Na⁺ or Li⁺, and is unable to catalyze all three modes of melibiose active transport coupled either H⁺, Na⁺, or Li⁺ as shown in the Figure 1. Remarkably, this mutant can ferment melibiose (means a downhill transport at high extracellular melibiose concentration) and catalyze melibiose exchange at rates similar to the wild-type MelB_{St} as reported previously. Similar conclusion was also obtained from the D59C MelB_{Ec} as discussed in this manuscript.

2. The authors need to provide composite omit maps to confirm the presence of each ligand in the structures, which are important to avoid any model bias. Only 2Fo-*fc* maps are provided in Fig. 3 and 4.

The composite omit maps from both structures were used for verification during modeling, which was not included in the first version. Now we have provided as shown in **SI Figure S6** and discussed in the text.

3. The authors describe the ligand-binding mechanism in great atomic details on pg. 8-10. They need to be careful given the modest resolution (3 angstroms) of the structures. However, no functional data support it. Can they confirm the binding mechanism by mutations?

There are strong electron densities between sugar and protein. In addition, the two ligand structures mutually support the specific galactosyl binding. MelB is an old protein that was discovered in 1965 with many mutagenesis results available as described in page 12 in the original version. In particular, the mutagenesis data on the essential sites for sugar binding in MelB_{St} have been published previously. Here is the quote:

“Mutations on most of these positions largely reduce the binding or transport in MelB_{St}⁽⁷⁾ or MelB_{Ec}, and some affected affinity to both sugar and cation^(34, 47-50). The structure was also strongly supported by the Fourier-transform infrared spectroscopy (FTIR) studies in MelB_{Ec}, where the mutant on positions-19 or -124 lost sugar-specific FTIR signals and D124C also largely reduced the Na⁺-specific signals⁽³⁴⁾.”

My lab has completed a complete Cys-scanning mutagenesis of MelB_{St}, which supports this binding site. With 3.5 Å distance to the sugar, Cys mutation on all contacting residues abolished melibiose active transport with regard to initial rate and steady state level of accumulation. We will publish the complete Cys-scanning data in the separate article.

I agree with your comment, and careful descriptions are indeed needed. It is challenging, but we need a parameter to describe the structure. I added a sentence to emphasize this caution in pages 8-9 as quoted here:

“The distance measurement will be useful to describe the relative orientation and define the hydrogen-bonding interactions.”

4. It is a bit surprising that no structure comparison is provided between the apo and ligand-bound forms. Any conformational changes between these two conformations? Any conformational changes induced by ligand binding in the substrate-binding pocket? This information is very important to understand the transport mechanism.

Thanks for this review's comments. Crystallization of MelB was very challenging and took decades from multiple laboratories worldwide. Myself and my lab substantially spent more than 7 years, and fortunately got the first crystal structure of WT MelB_{St}. We could only obtain a dataset diffracting to a level beyond 4 Å from a crystal with pseudo-translation, which prevented refinement. We were not able to model ambiguous densities in the cavity but described that melibiose was presented in the crystallization trials. We should not treat it as an apo state. It is not meaningful to compare with the first structure, which was not completely refined.

5. What is the purpose to use both nitrophenyl-galactoside and dodecyl-melibioside? Both structures are nearly identical and have similar binding conformation. No much more information from these duplicate works.

I am sorry that we do not agree with these comments. The DDMB binding is very useful and meaningful. We are not only interested in the protein structure but also focused on sugar binding specificity. As you mentioned that the resolution is at a modest level, the β -glycosidic bond on C1' position of glucosyl moiety can function as a marker for the verification of the orientation of the specific galactosyl moiety in the sugar specificity determinant pocket. The long tail of DDMB was not resolved, which positively added on the notion that non-galactosyl moiety does not have much specific interactions with MelB. This feature allows MelB to recognize a structurally and chemically unrelated ligands if containing the specific galactosyl moiety.

Minor concerns:

1. In the abstract, the authors state that “the conserved cation-binding pocket is also assigned”. How to assign the cation-binding pocket if they did not see any cation in the structure giving the fact that the D59C mutation has abolished cation binding?

I am sorry for this oversight. Now we have changed the “assigned” to “proposed”. There are bunch of available data supporting the direct role of D55 and D59 in the

Na⁺ binding in MelB. While Na⁺ is not presented, the immediate surrounding near both negatively charged positions is expected to be the cation site to host H⁺, Li⁺, or Na⁺. The structure shows that this site is directly linked to the C6-OH on the specific galactosyl moiety, and this supports for our findings on positive cooperativity between the two sites. Certainly, we will need a structure with bound Na⁺ or Li⁺ to confirm the cation site.

2. The manuscript needs substantial revision. The authors should focus on their own work instead of discussing too much about other human transporters. For example, the first paragraph can be summarized in a couple of sentences. I understand that MelB belongs to the MFS_2 family, which contains several human transporters. However, lipid transporter MFSD2A may have a completely different mechanism. The authors need to be careful to avoid overstatement. Also need to improve the English. Many typos and grammatical errors make it hard to follow.

Thanks for the comments. We have extensively revised the manuscript. The MelB human transporters, especially the lipid transport shared many functionally conserved residues, and the reviewer #2 demanded more discussions on MFSD2A and even for a threading model. While lipids translocation path may differ from sugar transport, the data collected so far show that cation sites are conserved. The knowledge collected from MelB in the past decades have been successfully transferred for our better understanding of these human homologues. We kept these comparisons limited and did not make overstatements.

Reviewer #2 (Remarks to the Author):

Overall, the data analysis and biochemical studies support the main conclusions of this paper. This work is timely and important to multiple fields of research that span interests in transporters and medicine and can be used as a roadmap to better understand how other clinically relevant MFS transporters work. In this regard, this work could be improved by addressing the following comments/suggestions.

Thank this reviewer for your recognition of the importance of our studies.

1) Does the fact that the D95C structure capture sugar ligand binding indicate that it is possible that sugar binding can precede cation binding in the transport cycle?

This is a very interesting question. Influenced by the knowledge generated from LacY, where the sugar binding can only be detected with protonated LacY, we thought that the binding is ordered even with purified protein samples. Because the pKa in LacY is so high (>10.5), it is not suitable for most conventional techniques to study the apo LacY (deprotonated with no sugar). In MelB, the pKa is <6.5, we found that the apo MelB (no H⁺, Na⁺, Li⁺, sugar) is stable and is able to bind either substrate (sugar and cation). We also found that positive cooperativity in the co-substrates binding to MelB exists across all three types of coupling cations with sugar. While

sugar binding can be independent, during the transport cycle, the outward-open apo MelB should bind cation first and sugar second; i.e., the long-standing stepped-binding model generated from LacY and MelB is still valid with regard to the transport cycle. This is because apo MelB exhibits very low affinity for sugar (10 mM) but with 20-fold higher affinity for Na⁺ (0.5 mM); in addition, the availability for Na⁺ should be greater than that for sugar in most environments.

2) DDMB exhibited 4X higher affinity than a-NPG in the presence of sodium. The authors reveal a “non-specific” hydrophobic pocket. Does this pocket allow for increased binding affinity for DDMB. The reason for this question is that the authors mention Mfsd2a which is a lysolipid transporter and based on modelling against MelB has a much larger hydrophobic pocket than MelB. Mfsd2a can also transport LPCs with 12 carbon acyl chains (as DDMB contains). Is this pocket in MelB and Mfsd2a in similar regions of the protein and composed of similar transmembrane domains?

We know that the non-galactosyl moiety increases the binding affinity, especially by adding hydrophobic moieties. This is observed in both LacY and MelB. During the sugar recognition that must be completed in open states, the structurally and chemically diverse non-specific non-galactosyl moieties can be hosted by the large non-specific binding pocket (we did not call it as non-specific hydrophobic pocket). When protein conformation is shifted into the occluded state, it is expected to have more non-specific interaction between the substrate and the protein; we have not obtained a structure to verify these ideas yet.

Based on our knowledge, the functionally important positions in Mfsd2a and MelB are structurally conserved, especially on the cation site. The conservation analysis support both proteins have a similar fold. I would say that the pocket in both proteins should be structurally conserved shared by similar transmembrane segments.

3) Does the PDB 4m64 structure and the other structure (partially occluded from the Guan lab, Nat Comm paper) in comparison to these new structures (apo vs holo) reveal any subtle conformational changes in the ligand binding pocket?

We addressed this question to the reviewer #1's comment (#4). Melibiose was presented in the crystallization trials but with no clear density for modelling. We should not treat it as an apo state. It is not meaningful to compare with the first structure which was not completely refined.

4) Is the S166 residue of human Mfsd2a which is mutated to S166L in familial autosomal microcephaly 15 homologous to S153 in MelB? The reason for this question is that there is little understanding of the role of S166 in Mfsd2a, and it might be possible to answer this question with these new structures.

The homologue position of S166 of MFSD2A in MelB is Trp128, not S153. Trp128 is located within the sugar specificity pocket donating H-bond to sugar and D124. In MFSD2A, I guess that the function of this region should be conserved. If so, Ser116 should be one important contributor to the specificity pocket, and S166L mutation may affect lipid recognition by losing possible H-bond interaction(s) with lipid head group. A brief discussion was added in page 15.

5) Can the authors comment on how they predict the cation transport path might look like for MelB and why Mfsd2a is strictly sodium dependent while MelB is not. Again, these questions are intended to improve the impact of this work and can be addressed with modelling or discussion.

We think that the cation translocation path for both proteins should be similar; i.e., it binds from or is released into the solvent-filled cavity, and diffuses away when a vestibule is open.

I do not know how much data support that “Mfsd2a is strictly sodium dependent”. The D92A mutant behaves as the WT based the results from incorporation into PC. What does that mean? The Na⁺ stimulation in MFSD2A is not so high even in WT. It is still challenging to figure out the metal binding in the absence of a structure. Sorry, I am not able to address this question. However, we did homologue modeling as suggested, and presented the outward-facing model of human MFSD2A and compared its cation site with MelB as shown in figure 5 c and d.

6) In the results section on ligand binding, can the authors comment on the functional testing of the residues that have been identified to coordinate sugar binding? I believe many of these residues have been tested for function in the authors’ previous studies and would be helpful to readers not familiar with MelB to point this out.

We have published mutagenesis data for these positions previously, and also mentioned the results in page 13. This manuscript has become too long to discuss the mutational effect on each position; however, I added few sentences to point out that in the results section. By the way, we have completed a Cys scanning mutagenesis for the full-length MelB_{st}, and more data will be available soon. Cys mutation on the major binding residues abolished transport completely.

7) It would be more appealing to the broader transporter community to modify the final sentence in the abstract as “These key structural findings lay a strong basis for understanding cooperative binding and symport mechanisms for sodium-dependent MFS transporters, including eukaryotic transporters such as Mfsd2a.”

Thanks for this comment. We are more than happy to modify this conclusive sentence to “These key structural findings lay a strong basis for our understanding on cooperative binding and symport mechanisms in Na⁺-coupled MFS transporters, including eukaryotic transporters such as MFSD2A.”

Reviewer #3 (Remarks to the Author):

The capability of using both Na⁺ and H⁺ to drive the transport makes MelB unique among the MFS. MelB has become also recently relevant due to its homology to some human secondary transporters: MFSD2A and MFSD13a.

MelB was extensively studied, mainly in the 80s and 90s, arguably being, after the Lactose permease, the best-understood secondary transporter.

In this occasion, Guan and Hariharan present two X-ray structures of a MelB mutant (D59C) bound to two high-affinity melibiose analogues: NPG and the detergent DDMB. Although this mutant lacks both Na⁺ and H⁺ binding, it shows for the first time how the galactoside ring of melibiose binds to the protein, in what the author call the galactoside pocket. It also shows a larger nonspecific binding pocket, which explains why MelB binds not only disaccharides but also trisaccharides or detergent molecules as DDMB. The results fits well with what was known about which are the important residues for melibiose binding, but with a level of detail that only structural data at atomic resolution can achieve. Therefore, it is a important step further in understanding MelB and related transporters. The paper is well written and well-illustrated by the included figures.

Thank you.

1. In the Results, the author switch from [3H]melibiose transport assays to ITC and fluorescence measurements without any clarification about the state of the protein. For the [3H]melibiose transport assays it would be better to specify that those are performed in whole-cells (to my understanding), while the rest of experiments (ITC, FRET, CD) are conducted in detergent solubilized MelB. This is an important clarification, as being a membrane protein without any further explanation readers might jump into the assumption that all experiments are done for MelB in lipidic membranes.

Thanks for this comment. It is important to clarify the experimental conditions. We have this information in the figure legends, but they should present in main text. Now these important details have been added in the text and figure.

2. The titration experiments of DDMB, DDM and UDM using FRET casts several questions. Regarding DDMB it is said that its concentration is below the cmc, to keep it monomeric. However, the cmc is defined when the only component in a solution is water, and will be affected by the presence of other detergents. It is very likely that the DDMB, even if below the cmc, will be incorporated in the UDM micelles that solubilize MelB. Another issue is the use of DDM and UDM as a negative control, showing that the binding of DDMB is driven by its melibiosyl moiety and did not occur when changed to a maltosyl. But the authors do not explain how these experiments were conducted.

Thanks for this comment. We shared your concern. We have clearly stated this concern with regard to the concentration issue in the results section of the previous version: “Since both monomer and micelles co-exist in the concentrated solutions used for the titration...”

Now we further stressed this concern in page 7 highlighted in yellow. Even DDMB affinity underestimated, it is the highest affinity ligand among all tested galactosides to MelB_{st}. **Importantly, use of this high-affinity substrate further confirms the uncoupling property with this mutant, as well as defines the positive cooperativity based on the specific galactosyl moiety with selected cations.**

As described in the text, all the FRET tests were carried out with purified MelB_{st} at 1 μM in 20 mM Tris-HCl, pH 7.5, 100 mM CholCl, 10% glycerol, 0.03% UDM in the absence or presence 100 mM NaCl. Yes, it must be sugar free for the detection of

D2G binding.

If DDM binding was tested for MelB solubilized in UDM, it is reasonable not to detect any binding, as any potential binding sites would be occupied by the UDM in excess. I can only guess that when testing DDM and UDM binding MelB was solubilized in a detergent lacking a sugar moiety.

We do not agree with this argument. If UDM occupied the sugar binding site, we should not detect the Trp to D2G FRET.

3. The experiments testing the thermoestability fo MelB WT and D59C are disturbing. They show how the ellipticity at 220 nm, related to the helicity, drops to almost zero above a certain temperature, indication that MelB loses its helical structure at high temperature. This is nonsense, as helical membrane proteins remain helical even when denaturated at high temperature. The “unfolding” of membrane proteins usually involves aggregation of loops, and rearrangement in the transmembrane helices, but without or very small changes in helicity. My only guest is that the CD signal goes to zero because the protein precipitates. In any case, the authors should include the CD spectrum at temperature above the T_m , and UV-vis or fluorescence spectra below and above the T_m , to confirm that the protein remains in solution.

We partially agree with this comment. Thermo-denaturation is a mixed process; some hydrophobic transmembrane helices may remain helical components even at aggregations, but not all helices in MelB is hydrophobic, especially these in loop and tail as well as forming the hydrophilic cavity. Our equipment currently does not have the capability to monitor the UV-vis or fluorescence spectra. However, we added the spectra traced for both WT and the mutant at each temperature. We admitted that the original statement is not precise and now we changed the denaturation profile to “Unfolded fractions” and modified the definition on T_m . “The ellipticity at 210 nm is decreased along with temperature increase reflecting helical unwinding, partial unwinding, and protein precipitations out of the solutions. The T_m values were determined as the temperature leading to the half maximal increase in the unfold fractions including precipitations by fitting the data using the Jasco Thermal Denaturation Multi Analysis Module.” Now we have only stressed the difference between the mutant and the WT. Thank you very much for improving our knowledge on this issue.

4. Somewhere in the MS appears “The373”. Thr373?

Sorry. This mis-spelling has been corrected now.

5. In the “proposed cation-binding pocket” the author, talking about Lys377, they say: “It could switch between a salt-bridge with Asp55 and a H-bond with C6-OH on the sugar.” This possibility is not supported by their data, but has been shown to be possible in MD simulations (Fuerst et al (2015) JBC). To my knowledge, the above work is the only one displaying how the cation-binding pocket of MelB could look like, so it should be taken into account in this section.

The structure reveals K377 is at 3.5 Å distance to D55 and 4.6 Å to C6-OH group. Since there is no density beyond Cε position, when I modeled the side chain, a rotamer

of Z nitrogen can be placed towards both D55 and sugar C6-OH. I believe what you meant that there is no Na⁺ binding configuration known in MelB and the MD modelling gave the first clue. I have acknowledged this article now.

6. At the end of the Results the authors mention the MFSD2A and MFSD2B transporters, giving a reference to Fig. S5, but the correct figure is Fig. S6.

It has been corrected.

7. All the characterization of the binding properties of MelB are done in detergent solubilized samples in the presence of 10% glycerol. Why glycerol was added? Why so much? MelB is a stable protein even in detergent, so I cannot believe it was necessary to add glycerol to make the ITC or FRET possible. Is there any other reason?

As you know, MelB_{St}, not MelB_{Ec}, is stable and functional in UDM. We added 10% glycerol mainly for storage purpose at – 80 C, and it may also serve as cryoprotectant to reduce x-ray damage of MelB crystals. Since we do not see negative effect or interference on other analysis, and we kept the sample conditions for crystallization unchanged for all other assays.

8. In the FRET experiments MelB is not only in the presence of 10% glycerol but also in the presence of 100 mM CholCl. What is CholCl? Why was it added?

We use CholCl to maintain same ionic strength. It seems that it is not really needed.

9. At the end of the “Methods” there is a sentence I cannot understand: “The first met was not present as also shown by N-terminal sequencing analyses with MelBSt (Alphalyse Inc. CA).”

The first position for methionine is not presented in MelBSt. The sentence has been revised now. “The first position for methionine was not present and likely processed post-translationally, which was verified by N-terminal sequencing analyses with MelB_{St} protein samples”

REVIEWERS' COMMENTS:

Reviewer #1 (Remarks to the Author):

The authors have addressed most of my questions. They have also provided composite omit maps for both structures.

My only concern remained is the weak electron density of DDMB. Based on Figure S6, the density of alpha-NPG was contoured at 1.5 sigma, but the map of DDMB was presented at a very low level (0.8 sigma). The authors need to be cautious to interpret such a weak density. If the density of DDMB is weak, the authors may either consider removing it or discuss it very carefully with support from other data.

Reviewer #2 (Remarks to the Author):

This revised version by Guan and co-workers is substantially improved by adding new analyses and clearer interpretations of some of the data. The overall findings of identifying the sugar binding residues of MelB is important work and adds to our understanding of substrate recognition by MelB and potentially other sodium dependent MFS transporters. This reviewer does not have further concerns.

Reviewer #3 (Remarks to the Author):

I have read with interested the comments from the rest of the reviewers as well as the reply from the authors. As often happens theirs work has clearly improved, in my opinion, during the revision. From my side I do not have any more comments or corrections, and I hope that the authors can obtain structures with better resolution in the future, confirming the here presented sugar binding site and revealing the cation binding site.